# Association between post-stroke depression and functional outcomes: A systematic review

Nipaporn Butsing[1]*, Jaclene A. Zauszniewski[2], Suebsarn Ruksakulpiwat[3], Mary T. Quinn Griffin[2], Atsadaporn Niyomyart[1]*

1 Ramathibodi School of Nursing, Faculty of Medicine Ramathibodi Hospital, Mahidol University, Bangkok, Thailand, 2 Frances Payne Bolton School of Nursing, Case Western Reserve University, Cleveland, Ohio, United States of America, 3 Department of Medical Nursing, Faculty of Nursing, Mahidol University, Bangkok, Thailand

* nipaporn.but@mahidol.edu (NB); atsadaporn.niy@mahidol.ac.th (AN)

## Abstract

### Background

Post-stroke depression (PSD) is a frequent problem in stroke patients, affecting their rehabilitation process and functional outcomes. Several studies have investigated the relationship between PSD and functional outcomes, but the results have been inconsistent.

### Objective

This systematic review of non-experimental studies aims to investigate the prevalence of post-stroke depression and the association between post-stroke depression and functional outcomes.

### Method

A search of PubMed, MEDLINE, Web of Science, and CINAHL Plus with Full Text was carried out from inception until January 2024. The literature was screened using the Preferred Reporting Items for Systematic Reviews and Meta-Analyses (PRISMA) guidelines, with relevant papers included. We extracted data from non-experimental studies that examined associations between PSD and functional outcomes. The Joanna Briggs Institute for systematic reviews was used for critical appraisal.

### Results

Twenty-one studies met the study criteria, including sixteen cohort studies, four cross-sectional studies, and one case-control study. PSD prevalences ranged from 12.2% to 32.2% in the first two weeks, 17.9 to 35.5% in the first month, and 10.4% to 32.0% in the third month following a stroke. Functional outcomes were evaluated in four domains: degree of dependence, basic activity of daily living, instrumental activity of daily living, and physical and cognitive function. Significant associations between PSD and functional outcomes were identified after controlling potential factors such as age, comorbidities, and stroke severity. PSD had negative associations with functional outcomes in all four measure domains from

**Data Availability Statement:** All relevant data are within the manuscript and its Supporting Information files.

**Funding:** The author(s) received no specific funding for this work.

**Competing interests:** The authors have declared that no competing interests exist.

one month to five years after a stroke. Depression treatment showed positive results on functional outcomes in stroke patients.

## Conclusion

PSD prevalence was high in the first three months after stroke. PSD is significantly associated with poor functional outcomes. PSD assessment and management should be performed on a frequent basis in the early stages of stroke to achieve the best possible functional recovery.

## Introduction

Stroke continues to be one of the world's leading causes of death and disability [1, 2]. The recent Global Burden of Disease (GBD) study estimates that stroke caused 160.4 million disability-adjusted life-years (DALYs) lost in 2021 [2], and age-adjusted death rate of stroke was 87.4 per 100,000 population [3]. Additionally, there were 12.2 million incident cases of stroke and 101 million prevalent cases of stroke globally in 2019 [4]. Furthermore, as the population ages, more people are experiencing high systolic blood pressure, high body-mass index, high fasting plasma glucose, ambient particulate matter pollution, smoking, high sodium consumption, and high LDL cholesterol [4]. These could be contributing factors to the rising stroke burden.

Neurological deficits from stroke include physical, cognitive, or psychological problems, depending on the severity and location of stroke lesion [5, 6]. However, stroke patients commonly experience neurological impairment and functional dependence in activities of daily living (ADL) [7–9]. The impact of stroke on functional outcomes could range from basic ADL to instrumental ADL, working ability, and social involvement. Several instruments have been used to measure the functional outcomes of patients with stroke [10]. Commonly used instruments for measuring functional outcomes in patients with stroke include 1) degree of dependence measures, such as modified Rankin Scale; 2) basic activity of daily living measures, like the Barthel index (BI), and the modified Barthel index (mBI); 3) instrumental activity of daily living measures, such as the instrumental activities of daily living (IADL), and the Frenchay Activities Index (FAI); and 4) physical and cognitive function measure, like the Functional Independence Measure (FIM) [10]. The functional outcomes are essential indicators of stroke rehabilitation and recovery. Typically, patients' functional status spontaneously recovers over six months after a stroke [11, 12]. However, the most significant and rapid recovery is regained during the first few months after stroke onset [12]. Moreover, recovery from a stroke depends on the lesion and severity of the stroke [6, 12], as well as individual factors such as age, gender, comorbidities, smoking, drinking, and depression [13–19].

Depression is a frequent condition among patients after experiencing a stroke. Post-stroke depression (PSD) is often caused by biomedical changes in the brain resulting from brain injury, as well as behavioral changes and social factors [20]. Patients who experience persistent sadness, anxiety, or apathy might not be able to feel positive emotions. Other depressive symptoms after a stroke include feelings of worthlessness, suicidal ideation, sleeping disturbances, diminished concentration, weight changes, psychomotor agitation, decreased energy, and fatigue [20]. Patients suffering from depression may become depressed and lose interest in rehabilitation and social interaction. Depressive symptoms usually occur within the first three months after a stroke, or they may occur at any time [14, 15, 18–21]. Depression affects one in

three stroke patients in the first year after a stroke [22]. Depression may make the rehabilitation process more difficult, resulting in poor recovery and functional outcomes [14, 15, 17, 20, 21].

Despite the large number of studies published on the association between post-stroke depression and functional outcomes after a stroke, the findings varied according to the study population, study design, PSD measurements, functional outcome measurements, and confounding variables. The purpose of this systematic review was to analyze how depression affects functional outcomes after a stroke. We identified and considered different factors in this systematic review when extracting the results. More specifically, themes were extracted from the main findings of the included studies throughout the data synthesis process by examining the similarities and differences of the main findings according to the domains of functional outcome measurements. Additionally, PSD measurements and study design issues were used to summarize PSD prevalences. This systematic review is expected to provide an up-to-date and comprehensive summary of PSD prevalence and its impact on functional outcomes.

## Methods

The methods used in this systematic review consisted of a search strategy, selection of studies, data extraction, and quality assessment of included studies. The review protocol was registered in the International Prospective Register of Systematic Reviews (PROSPERO) under the registration number CRD42024518850. Ethical approval was not required due to no human subjects being involved in this study.

### Search strategy

The Preferred Reporting Items for Systematic and Meta-Analysis (PRISMA) guidelines [23] were used to demonstrate the identification, screening, exclusion, and inclusion flow diagram (S1 and S2 Appendices). A systematic search was conducted using four electronic databases: PubMed, MEDLINE, Web of Science, and CINAHL Plus with Full Text on January 17, 2024, to identify preliminary studies reporting associations between post-stroke depression and functional outcomes in patients with stroke published between January 2013 and December 2023. In order to find relevant research, the PECO framework [24] was used in the following cases: P: Population = Stroke patients; E: Exposure = Depression; C: Comparison = None; and O: Outcome = Functional outcomes. The researcher combined the following search terms: (Stroke OR cerebrovascular accident OR cerebrovascular disease OR cerebrovascular disorders OR cerebral ischemia OR brain ischemia OR cerebral infarction OR ischemic stroke OR Embolic stroke OR hemorrhagic stroke OR Cerebral Hemorrhage OR intracranial hemorrhage, hypertensive OR post-stroke) AND (Depression OR Depressive Symptom* OR Emotional Depression OR Depressive Symptom* OR Unhappiness OR Sadness) AND (Functional Outcome OR Functional Recovery OR Functional Improvement OR Functional Impairment OR Functional Status OR Physical Functional Performance OR Disability OR Neurological Deficit OR Disabled Persons) using Boolean/Phrases. All references identified were stored in EndNote 20. The detailed search strategy is shown in S1 Table.

### Selection of studies

Non-experimental studies or observational analytic studies included cross-sectional, case-control, and cohort studies were selected for this review [25]. The details of the inclusion and exclusion criteria are presented in Table 1. Titles and abstracts were screened for eligible studies. Afterward, the full text was assessed to decide whether it was relevant. Inclusion criteria were implemented to guarantee that only studies considered relevant to the study objective

**Table 1. Study inclusion and exclusion criteria.**

| Inclusion Criteria | Exclusion Criteria |
|---|---|
| • The study included adults aged 18 years or older.<br>• Original observational analytic studies<br>• Studies reported functional outcomes of stroke, including ischemic stroke and hemorrhagic stroke.<br>• The exposure is defined as depression.<br>• Described in the English language. | • The study did not include the population of interest or concerned animal subjects.<br>• Pre-stroke depression<br>• Conference proceedings, abstracts, review articles, case study, case report, meta-analyses, theoretical papers, pilot studies, protocols, dissertations, letters to the editor, opinion (viewpoint), statement papers, government documents, protocol papers, or working papers. |

were included. Similarly, exclusion criteria were used to exclude literature that was not relevant to the review.

## Assessment of methodological quality

The methodology quality of each study was assessed by three authors using the Joanna Briggs Institute (JBI) Critical Appraisal Checklist designed for systematic review included JBI Critical Appraisal Checklist for cohort studies, JBI Critical Appraisal Checklist for analytical cross-sectional studies, and JBI Critical Appraisal Checklist for case control studies [26, 27]. Any disagreement among the three authors was resolved by discussion that resulted in a consensus. Studies with a JBI score higher than 70% were categorized as high quality, moderate quality if scored between 50% and 70%, and low quality if scored lower than 50% [26].

## Data extraction and data analysis

The summarized data extraction for each study included its reference, year, country, study aims, study design, number of participants (N), study population, age (years), measures, time points of assessment, results (S2 Table). Data analysis was carried out by collecting and synthesizing information on general characteristics, methodologies, assessment of depression and functional outcomes, and confounding factors.

## Data synthesis

The results of the included studies were compiled using the convergent integrated analytic approach recommended by the Joanna Briggs Institute (JBI) for systematic reviews [26, 27]. The prevalence of depression after stroke was reported based on study designs—cohort study and cross-sectional study. Data synthesis was done narratively since there was heterogeneity in included study characteristics. Measures of functional outcomes were used to identify themes relating to the association between post-stroke depression and functional outcomes. The tabular approach was utilized to pool and categorize data based on similarities and differences in meaning to create a group of integrated findings for data synthesis. The main findings are based on four functional outcome measure domains: 1) degree of dependence, 2) basic activities of daily living, 3) instrumental activities of daily living, and 4) physical and cognitive function. Moreover, sub-themes were then abstracted according to the more specific target of related results as needed [26, 27].

## Results

## Search results

Following the PRISMA guidelines [23], a total of 2,006 were initially identified from PubMed and MEDLINE (n = 114), Web of Science (n = 1,581), and CINAHL Plus with Full Text

(n = 311). A total of 304 duplicate studies were identified and removed. Another 135 articles were removed due to their ineligibility based on the type of study. The remaining papers were then filtered using their titles and abstracts according to inclusion and exclusion criteria. At this stage, 33 articles were eligible for full-text retrieval since 1,534 were removed from the review because they did not meet the inclusion criteria. However, the full text of 12 articles could not be retrieved. Consequently, the quality assessment comprised 21 studies (Fig 1).

## Characteristics of the included studies

Table 2 shows characteristics of 21 included studies from 2013 to 2023. The majority (66.67%) were published in the recent five years (2019–2023); 2019 and 2021 had the highest number of publications (n = 5 studies, 23.81% for each year). Five studies were conducted in China (23.81%). Two studies (9.52%) were conducted each in Korea and Japan. The USA, Canada, UK, Sweden, Poland, Italy, Germany, Iran, India, Turkey, Ethiopia, and Nigeria were among the countries where only a single study (4.76%) was carried out. Most of the study designs were cohort studies (n = 16, 76.19%), followed by cross-sectional studies (n = 4, 19.05%), and one was a case-control study (4.76%). Ischemic stroke (n = 10, 47.62%) accounted for the majority of the target population in the included studies, followed by acute ischemic stroke (n = 8, 38.10%) and hemorrhagic stroke (n = 8, 38.10%). Five studies (23.81%) included sample sizes larger than 500. There were ≤100, > 100 to 200, and > 200 to 300 participants in each of the four studies (19.05%). Two studies (9.52%) included > 300 to 400 and > 400 to 500 participants. The validated scales used to measure depression following a stroke were the Hamilton

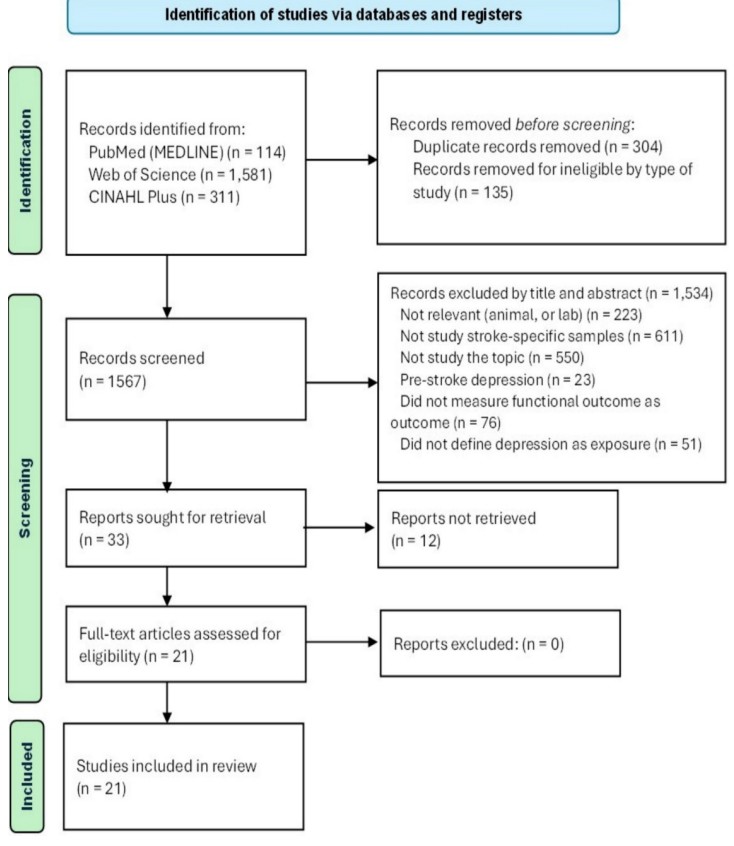

**Fig 1. PRISMA flow chart [23].**

**Table 2. The characteristics of included studies.**

| Characteristics | Number* | Percentage |
|---|---|---|
| **Publication year** | | |
| 2023 | 1 | 4.76 |
| 2022 | 1 | 4.76 |
| 2021 | 5 | 23.81 |
| 2020 | 2 | 9.52 |
| 2019 | 5 | 23.81 |
| 2018 | 2 | 9.52 |
| 2017 | 2 | 9.52 |
| 2016 | 1 | 4.76 |
| 2015 | 2 | 9.52 |
| 2014 | 0 | 0.00 |
| 2013 | 0 | 0.00 |
| **Country** | | |
| China | 5 | 23.81 |
| Korea | 2 | 9.52 |
| Japan | 2 | 9.52 |
| UK | 1 | 4.76 |
| USA | 1 | 4.76 |
| Canada | 1 | 4.76 |
| Sweden | 1 | 4.76 |
| Poland | 1 | 4.76 |
| Italy | 1 | 4.76 |
| Germany | 1 | 4.76 |
| Iran | 1 | 4.76 |
| India | 1 | 4.76 |
| Turkey | 1 | 4.76 |
| Ethiopia | 1 | 4.76 |
| Nigeria | 1 | 4.76 |
| **Study design** | | |
| Cohort study | 16 | 76.19 |
| Cross-sectional study | 4 | 19.05 |
| Case-control study | 1 | 4.76 |
| **Target population** | | |
| Acute ischemic stroke | 8 | 38.10 |
| Ischemic stroke | 10 | 47.62 |
| Hemorrhagic stroke | 8 | 38.10 |
| Transient ischemic attack | 1 | 4.76 |
| Unspecified stroke | 3 | 14.29 |
| Others | 2 | 9.52 |
| **Sample size (n)** | | |
| ≤ 100 | 4 | 19.05 |
| > 100–200 | 4 | 19.05 |
| > 200–300 | 4 | 19.05 |
| > 300–400 | 2 | 9.52 |
| > 400–500 | 2 | 9.52 |
| > 500 | 5 | 23.81 |
| **Depression measurement** | | |

(*Continued*)

**Table 2.** (Continued)

| Characteristics | Number* | Percentage |
|---|---|---|
| The Center for Epidemiologic Studies Depression Scale (CES-D) | 1 | 4.76 |
| Patient Health Questionnaire-9 (PHQ-9) | 3 | 14.29 |
| Patient Health Questionnaire-8 (PHQ-8) | 1 | 4.76 |
| Patient Health Questionnaire-2 (PHQ-2) | 1 | 4.76 |
| Hospital Anxiety and Depression Scale (HADS) | 3 | 14.29 |
| Hamilton Depression Rating Scale (HDRS) | 5 | 23.81 |
| Beck Depression Inventory (BDI) | 3 | 14.29 |
| Geriatric Depression Scale (GDS) | 1 | 4.76 |
| Self-Rating Depression Scale (SDS) | 2 | 9.52 |
| Mini International Neuropsychiatric Interview (MINI) | 1 | 4.76 |
| Montgomery–Åsberg Depression Rating Scale (MADRS-J) | 1 | 4.76 |
| Cornell Scale for Depression (CDS) | 1 | 4.76 |
| Stroke Aphasia Depression Questionnaire (SADQ) | 1 | 4.76 |
| **Functional outcome measurement** | | |
| Modified Rankin Scale (mRS) | 8 | 38.10 |
| Barthel index (BI) | 8 | 38.10 |
| Modified Barthel Index (mBI) | 2 | 9.52 |
| Lawton Instrumental Activity Daily of Living (IADL) | 2 | 9.52 |
| The Frenchay Activities Index (FAI) | 3 | 14.29 |
| Functional Independence Measure (FIM) | 3 | 14.29 |

Note: *one or more in one study

Depression Rating Scale (HDRS) (n = 5, 23.81%), Patient Health Questionnaire-9 (PHQ-9) (n = 3, 14.29%), Hospital Anxiety and Depression Scale (HADS) (n = 3, 14.29%), Beck Depression Inventory (BDI) (n = 3, 14.29%), Self-Rating Depression Scale (SDS) (n = 2, 9.52%). Additionally, each one study (4.76%) used The Center for Epidemiologic Studies Depression Scale (CES-D), Geriatric Depression Scale (GDS), Mini International Neuropsychiatric Interview (MINI), Cornell Scale for Depression (CDS), Montgomery–Åsberg Depression Rating Scale (MADRS-J), Stroke Aphasia Depression Questionnaire (SADQ), Patient Health Questionnaire-8 (PHQ-8), and Patient Health Questionnaire-2 (PHQ-2).

## Methodological quality assessment of the included studies

The JBI critical appraisal checklists for analytical cohort studies, cross-sectional studies, and case-control studies were used to assess the methodological quality of the included studies [26, 27]. The results show that content about methodological quality was clearly reported in the included studies (all above 50%), thus minimizing the risk of bias in study design, conduct, and analysis (S3 Table). Twenty (95.2%) studies were classified as high quality, and one (4.8%) study was classified as moderate quality. No paper with low quality was included in this review. However, because all 16 cohort study participants were enrolled following a stroke and the functional status was disclosed, they were not initially free of the outcomes [14–19, 21, 28–41]. In addition, three of the twenty-one included studies [32, 39, 40] did not specify the strategies used to address confounding factors, and two of the twenty-one studies failed to identify confounding factors [39, 40]. One cohort study was unclear about duration of follow-up, the reasons of loss follow-up, and the strategies for dealing with incomplete follow-up [33].

## Post-stroke depression prevalence

A total of 14 cohort studies reported the prevalence of depression after stroke [14–16, 18, 19, 21, 29, 32, 36–41]. Using subjective assessments such as the Mini International Neuropsychiatric Interview (MINI), the Cornell Scale for Depression (CDS), and the Self-Rating Depression Scale (SDS), three cohort studies found that the prevalence of depression among stroke survivors ranged from 12.2% to 32.2% between 1 and 2 weeks [15, 18, 36]. According to the objective assessment (Montgomery–Åsberg Depression Rating Scale, MADRS-J), there was a high prevalence of depression (68.4%) in the first few weeks [36].

Prevalence of depression one month after stroke was 17.9% to 35.5%, classified by Self-Rating Depression Scale (SDS) [36] and Hamilton Depression Rating Scale (HDRS) [14]. According to the Montgomery–Åsberg Depression Rating Scale, more than 50% of stroke patients were diagnosed with depression in the initial months following their stroke [35]. Four cohort studies reported depression prevalence at three months after stroke ranged from 10.4% to 32.0% identified by Patient Health Questionnaire-8 (PHQ-8) [21], Patient Health Questionnaire-9 (PHQ-9) [15], Hamilton Depression Rating Scale (HDRS) [41], and Hospital Anxiety and Depression Scale-Depression (HADS-D) [19].

The prevalence of depression at one year after stroke ranged from 7.13% to 9.28% assessed by Patient Health Questionnaire-8 (PHQ-8) [21] and at five years was 7.2% assessed by Hamilton Depression Rating Scale (HDRS) [14]. Moreover, two cross-sectional studies reported prevalence of depression among patients who had stroke more than 6 months and 1 year were about 43.3% [37] and 80.0% [29], respectively. The prevalence of depression after stroke varied by assessment time and measurements, as details shown in Table 3.

## Association between post-stroke depression and functional outcomes

Table 4 summarizes the research findings regarding the relationship between post-stroke depression and the functional outcome's theme based on functional outcomes measure domains. Scrutiny of Table 4 can be placed into four domains: 1) Degree of dependence (Modified Rankin Scale, mRS), 2) Basic activities of daily living (Barthel Index, BI; Modified Barthel Index, mBI), 3) Instrumental activities of daily living (Lawton Instrumental Activity Daily of Living, IADL; The Frenchay Activities Index, FAI), and 4) Physical and cognitive function (Functional Independence Measure, FIM). The results of the association between depression after a stroke and each functional outcome domain were observed differently by study designs, depression measurement, assessment time and confounding factor controlled.

**Degree of dependence.** The degree of dependence refers to a patient's disability in their daily activities. The modified Rankin Scale (mRS) is commonly used to measure the degree of dependence or disability in stroke patients [10]. The modified Rankin Scale was used in seven cohort studies [14, 15, 17, 21, 33, 40, 41]. These studies reported significant variation in terms of timing of assessment, depression assessment tools, and form of report degree of dependence. Therefore, these results are presented narratively. Six studies reported that depression after a stroke is a factor associated with a patient's degree of dependence. After adjusting for confounding variables, individuals with depressive symptoms after a stroke had a higher risk of poor functional outcome compared to those without depressive symptoms at three months (about 1.05 times [17] to 2.59 times [15]), twelve months (about 2.30 times [21] to 3.97 times [15]), and five years (about 1.80 times [41] to 2.28 times [14]).

However, the degree of stroke severity determined functional outcome improvement with time, whether with or without depression. According to the study by Kim et al., 2022 [33], they identified depression by GDS, and reported that depression had no significant effect on mRS changes in patients with moderate and severe stroke. Sharma et al., 2021 [40] carried out

**Table 3. Depression prevalence in patients with stroke.**

| Authors, Year, Ref # | Study design | Depression measures | Depression prevalence |
|---|---|---|---|
| Ayerbe et al., 2015 [19] | Cohort study | HADS-D | n = 418 (32.0%) at 3 months |
| Blomgren et al., 2019 [28] | Cross-sectional study | HADS-D | NA |
| El Husseini et al., 2017 [21] | Cohort study | PHQ-8 | Depression prevalence at 3 months and 12 months<br>n = 134 (9.28%) Persistent depression (3m+,12m+)<br>n = 126 (8.72%) Resolved depression (3m+, 12m-)<br>n = 103 (7.13%) Incident depression (3m-, 12m+) |
| Ezema et al., 2019 [29] | Cross-sectional study | HDRS | n = 56 (80.0%) had depression (>1 year of stroke)<br>n = 32 (57.14%) had severe depression (>1 year of stroke) |
| Ghaffari et al., 2021 [30] | Cross-sectional study | BDI | NA |
| Kang et al., 2018 [18] | Cohort study | MINI | n = 108 (25.5%) at 2 weeks |
| Kapoor et al., 2019 [31] | Cohort study | PHQ-2, CES-D | NA |
| Karaahmet et al., 2017 [32] | Cohort study | BDI | n = 49 (53%) at less than 6 months (not specified details) |
| Kim et al., 2022 [33] | Cohort study | GDS | NA |
| Li et al., 2019 [34] | Cohort study | HDRS | NA |
| Lopatkiewicz et al., 2021 [15] | Cohort study | PHQ-9 | Depression prevalence at 8 days and 3 months<br>n = 41 (12.23%) (8d+, 3m-)<br>n = 46 (13.73%) (8d-, 3m+)<br>n = 35 (10.45%) (8d+, 3m+) |
| Lv et al., 2021 [35] | Cohort study | SDS | NA |
| Matsuzaki et al., 2015 [36] | Cohort study | SDS, MADRS-J | 23.9% (SDS), 68.4% (MADRS-J) at 10 days<br>17.9% (SDS), 56.4% (MADRS-J) 4–6 weeks after admission |
| Mohammed et al., 2023 [37] | Cross-sectional study | PHQ-9 | n = 65 (43.3%) (>6 months) |
| Nakamori et al., 2020 [38] | Cohort study | PHQ-9 | n = 62 (28.8%) at less than 4 months (not specified details) |
| Paolucci et al., 2019 [39] | Case-control study | BDI | Case control (n = 280 depressed/n = 280 non-depressed) |
| Schöttke et al., 2020 [16] | Cohort study | CDS | n = 56 (32.2%)-7 days (T1)<br>n = 31 (36.9%)-3 years (T2) |
| Sharma et al., 2021 [40] | Cohort study | HADS-D, HDRS | n = 17 (57.7%) at less than 1 year (not specified details) |
| Wang et al., 2018 [17] | Cohort study | SADQ | NA |
| Yang et al., 2016 [41] | Cohort study | HDRS | n = 198 (22.2%) at 3 months |
| Zeng et al., 2021 [14] | Cohort study | HDRS | n = 154 (35.3%) at 1 month<br>n = 26 (7.2%) at 5 years |

Abbreviations: HADS-D, Hospital Anxiety and Depression Scale-Depression; HDRS, Hamilton Depression Rating Scale; BDI, Beck Depression Inventory; MINI, Mini International Neuropsychiatric Interview; CES-D, The Center for Epidemiologic Studies Depression Scale; PHQ-2, Patient Health Questionnaire-2; PHQ-8, Patient Health Questionnaire-8; PHQ-9, Patient Health Questionnaire-9; GDS, Geriatric Depression Scale; CDS, Cornell Scale for Depression; SDS, Self-Rating Depression Scale; MADRS-J, Montgomery–Åsberg Depression Rating Scale; SADQ, Stroke Aphasia Depression Questionnaire

Notes: n, number; d, days; m, months; "+", depression occurrence; "-", no depression occurrence.

another study on the effect of post-stroke depression on functional outcomes during inpatient rehabilitation. They found that both patients with and without depression had improved functional outcomes from admission to discharge. However, they did not find a significant effect of depression on mRS scores at the time of discharge compared with admission scores; nonparametric statistics were used to analyze the data without confounding control.

**Basic activities of daily living.** Basic activities of daily living (BADL) are fundamental skills related to personal care, including mobility and self-care tasks [42]. Eight included studies assessed BADL using Barthel Index (BI) [16, 18, 19, 29, 37, 40], and the modified Barthel Index (mBI) [35, 39]. These studies reported significant variation in study designs, timing of assessment, depression assessment tools (7 different tools), and form of report ADL. Therefore, studies within this group are reported narratively. Depression was found to have a significant impact on BADL in five cohort studies [16, 18, 19, 35, 40]. According to the study by

**Table 4. Results of association between post-stroke depression and functional outcomes.**

| Authors, Year, Ref # | Study design | Depression measures | Methodological quality | Functional outcome measure domains | | | | | | Controlling factors and remark |
|---|---|---|---|---|---|---|---|---|---|---|
| | | | | DEP | BADL | | IADL | | PhyCog | |
| | | | | Theme based on functional outcome measures | | | | | | |
| | | | | mRS | BI | mBI | IADL | FAI | FIM | |
| Ayerbe et al., 2015 [19] | Cohort study | HADS-D | High | | Y₁ N₂ | | | | | 1 = age, gender, ethnicity, stroke severity and past medical history of diabetes mellitus, hypertension, ischemic heart disease, heart failure or atrial fibrillation<br>2 = 1 + [physical disability, cognitive impairment, smoking habit, use of selective serotonin reuptake inhibitors, subjective perception, or social support at 3 months] |
| Blomgren et al., 2019 [28] | Cross-sectional study | HADS-D | High | | | | | N<br>Y*,₁<br>N** | | 1 = BNIS, gender, living alone, and severity of stroke (NIHSS)<br>* FAI-work/leisure<br>** FAI-domestic chores |
| El Husseini et al., 2017 [21] | Cohort study | PHQ-8 | High | Y₁ | | | | | | 1 = age |
| Ezema et al., 2019 [29] | Cross-sectional study | HDRS | High | | Y N₁ | | | | | 1 = age and duration of stroke |
| Ghaffari et al., 2021 [30] | Cross-sectional study | BDI | High | | | | Y₁ | | | 1 = age, TMT, and basic activities of daily living (BADL) performance |
| Kang et al., 2018 [18] | Cohort study | MINI | High | | Y₁ Y₂ | | | | | 1 = BI in the acute phase<br>2 = 1 + [age, previous depression and stroke, stroke hemisphere, and stroke location] |
| Kapoor et al., 2019 [31] | Cohort study | PHQ-2, CES-D | High | | | | | N*<br>Y**,₁ | | 1 = age, stroke severity, and cognitive symptoms<br>* CES-D<br>** PHQ-2 |
| Karaahmet et al., 2017 [32] | Cohort study | BDI | High | | | | | | Y | |
| Kim et al., 2022 [33] | Cohort study | GDS | Moderate | Y₁ N₂ | | | | | | 1 = MMSE, time, MMSE x time, age, sex, initial NIHSS, hypertension, diabetes mellitus, dyslipidemia, atrial fibrillation, and intravenous thrombolysis<br>2 = severity of stroke |
| Li et al., 2019 [34] | Cohort study | HDRS | High | | | | Y₁ | | | 1 = age, sex, NIHSS, and HDRS in the acute stage |
| Lopatkiewicz et al., 2021 [15] | Cohort study | PHQ-9 | High | Y*,₁ Y**,₂ | | | | | | 1 = age, NIHSS score, pre-stroke cognitive decline, NPI score, and delirium<br>2 = 1 + [hypertension, atrial fibrillation]<br>* Functional outcome at 3 months<br>** Functional outcome at 12 months |
| Lv et al., 2021 [35] | Cohort study | SDS | High | | | Y₁ | | | | 1 = age, stroke severity, cognition, and social support |
| Matsuzaki et al., 2015 [36] | Cohort study | SDS, MADRS-J | High | | | | | N*,₁<br>Y**,₁ | | 1 = gender, age, length of stay, FIM score on admission, MMSE, and apathy<br>* SDS<br>** MADRS-J |
| Mohammed et al., 2023 [37] | Cross-sectional study | PHQ-9 | High | | N₁ | | Y₁ | | | 1 = age, stroke duration, history of substance use, comorbid, aphasia, cognitive impairment, and initial NIHSS |
| Nakamori et al., 2020 [38] | Cohort study | PHQ-9 | High | | | | | | Y₁ | 1 = age, sex, solitude, comorbid diseases, NIHSS, stroke subtypes, location of lesion, MMSE, and FIM on admission |

*(Continued)*

**Table 4.** (Continued)

| Authors, Year, Ref # | Study design | Depression measures | Methodological quality | Functional outcome measure domains | | | | | | Controlling factors and remark |
|---|---|---|---|---|---|---|---|---|---|---|
| | | | | DEP | BADL | | IADL | | PhyCog | |
| | | | | Theme based on functional outcome measures | | | | | | |
| | | | | mRS | BI | mBI | IADL | FAI | FIM | |
| Paolucci et al., 2019 [39] | Case-control study | BDI | High | | | Y₁ | | | | 1 = Antidepressant drugs |
| Schöttke et al., 2020 [16] | Cohort study | CDS | High | | Y₁ | | | | | 1 = sex, age, and functional impairment on acute stroke |
| Sharma et al., 2021 [40] | Cohort study | HADS-D, HDRS | High | N*<br>N** | Y*<br>Y** | | | | | * HADS-D<br>** HDRS |
| Wang et al., 2018 [17] | Cohort study | SADQ | High | Y₁ | | | | | | 1 = sex, marital status, stroke type, aphasia at baseline, pulmonary infection at baseline, deep vein thrombolysis at baseline, and stroke severity on admission |
| Yang et al., 2016 [41] | Cohort study | HDRS | High | Y₁ | | | | | | 1 = age, sex, education, diabetes mellitus, cardiac disease, smoking, alcohol drinking, stroke history, NIHSS score at admission, cognitive impairment at 3 months, and stroke recurrence within 5 years |
| Zeng et al., 2021 [14] | Cohort study | HDRS | High | Y₁ | | | | | | 1 = age, sex, body mass index, history of smoking, and alcohol drinking, education years, baseline NIHSS score, baseline MMSE score, medical history (coronary heart disease, diabetes mellitus, hyperlipidemia, hypertension, stroke). |

Notes: "Y": Yes, there is significant association between post-stroke depression and functional outcome; "N": No significant association found between post-stroke depression and functional outcomes

Abbreviations: mRS, Modified Rankin Scale; FIM, Functional Independence Measure; FAI, The Frenchay Activities Index; BI, Barthel index; mBI, Modified Barthel Index; IADL, Lawton Instrumental Activity Daily of Living; HADS-D, Hospital Anxiety and Depression Scale-Depression; HDRS, Hamilton Depression Rating Scale; BDI, Beck Depression Inventory; MINI, Mini International Neuropsychiatric Interview; CES-D, The Center for Epidemiologic Studies Depression Scale; PHQ-2, Patient Health Questionnaire-2; PHQ-8, Patient Health Questionnaire-8; PHQ-9, Patient Health Questionnaire-9; GDS, Geriatric Depression Scale; CDS, Cornell Scale for Depression; SDS, Self-Rating Depression Scale; MADRS-J, Montgomery–Åsberg Depression Rating Scale; SADQ, Stroke Aphasia Depression Questionnaire; BNIS, Barrow Neurological Institute Screen for higher cerebral functions; NIHSS, National Institute of Health Stroke Scale; TMT, Trail Making Test; MMSE, Mini-Mental State Examination; NPI, Neuropsychiatric Inventory, DEP, Degree of dependence; BADL, Basic activities of daily living; IADL, Instrumental Activity Daily of Living; PhyCog, Physical and cognitive function

Ayerbe et al., 2015 [19], the prevalence of depression three months after stroke increased the risk of severe disability (BI = 0–14) and moderate disability (BI = 15–19) at three years after controlling confounding variables with RR 4.01 and 1.89, respectively. Another study in China found significant correlations between depression and disability (mBI ≤ 95) three years after the stroke, with an OR of 1.03 (95% CI, 1.01–1.06) after adjusting for age, stroke severity, cognition, and social support [35]. Two cohort studies revealed that depression during the acute phase of stroke significantly decreased functional ability in BADL at one year [18] ($\beta$ = -0.162) and three years [16] ($\beta$ = -16.47) when confounding factors were controlled.

However, two cross-sectional studies reported no association between depression and functional outcomes in BADL [29, 37]. Furthermore, one case-control study in Italy [39] reported that PSD patients who responded to antidepressant treatment had higher BI scores at discharge compared with those who did not respond to antidepressant treatment.

**Instrumental activities of daily living.** Instrumental activities of daily living (IADL) refer to more complex skills related to an individual's independent living in the community [42]. Five studies reported instrumental activities of daily living using Lawton instrumental activities

of daily living (IADL) [30, 34] and The Frenchay Activities Index (FAI) [28, 31, 37]. These studies reported significant variation in terms of study designs, timing of assessment, depression assessment tools, and form of report IADL. Therefore, these results are presented narratively. Two cohort studies reported that depression following a stroke was associated with functional outcomes in IADL scores [31, 34]. The study by Li et al., 2019 [34] demonstrated a significant association between depression scores during the acute phase of stroke and 3-month IADL after controlling for potential confounding factors. Another study by Kapoor et al., 2019 [31] found that depression had a negative association with IADL scores 2 to 3 years after stroke ($\beta$ = -2.41) when controlling for age, stroke severity, and cognitive function. According to three cross-sectional studies [28, 30, 37], post-stroke depression was associated with poor IADL. The study by Blomgren et al., 2019 [28] found that depression significantly affected FAI work/leisure for all stroke patients at seven years after stroke (OR = 1.10; 95% CI, 1.02–1.19) but it did not affect FAI domestic chores. Subgroup analysis, however, revealed that depression was significantly associated with both total FAI and FAI domestic chores after seven years for stroke patients without neurological deficits (NIHSS, 0) [28].

**Physical and cognitive function.**   Physical and cognitive function refers to an individual's functional ability to perform physical tasks, communication, and cognition [10]. The Functional Independence Measure (FIM) was used in three cohort studies to assess cognitive and physical function [32, 36, 38]. These three papers had significant variations in terms of study designs, timing of assessment, depression measures, and reporting of physical and cognitive function. Therefore, studies within this group were presented narratively. According to a prospective study in Turkey [32], stroke patients without depression had greater FIM changes from the time of admission to 1 month after discharge than stroke patients with depression (p<0.05). A retrospective cohort study in Japan [38] calculated the FIM gain score (FIM score at discharge–FIM score on admission). The average time spent in the rehabilitation ward was 82 days (SD = 43 days). After controlling other potential confounders, they found that the PHQ-9 score on admission was negatively associated with the FIM gain score ($\beta$ = -0.745) [38].

Moreover, another prospective cohort study conducted in Japan [36] measured depression after stroke using both a subjective scale (Self-rating depression scale: SDS) and objective scale (Montgomery–Åsberg Depression Rating Scale: MADRS-J), as well as functional outcomes using FIM on admission to a rehabilitation hospital and 4–6 weeks later. It was found that subjective SDS scores had no significant effect on stroke patients' FIM recovery. Nevertheless, depression determined by the objective scale (MADRS-J) showed a significant impact on the FIM recovery score after controlling confounding factors, including gender, age, length of stay, FIM score on admission, MMSE, and apathy [36].

## Discussion

Depression is believed to be caused by biological and psychological changes that occur following a stroke [43–45]. Stroke-related brain injury directly impacts neural pathways involved in mood regulation and depression [45, 46]. The results of this comprehensive analysis found that prevalence of depression following a stroke was documented in 14 cohort studies. According to standard subjective measures, the prevalence of post-stroke depression (PSD) in the first two weeks after a stroke ranged from 12.2 to 32.2% [15, 18, 36]. The PSD prevalence seemed to increase at one month (17.9 to 35.5%) [14, 36] and three months (10.4% to 32.0%) [15, 19, 21, 41], then decreased with the PSD prevalence at one year were 7.1% to 9.3% [21], and at five years was 7.2% [14].

Acute strokes are stressful events that cause abnormal neurotransmitter secretion, which might result in depression [47, 48]. Moreover, a high prevalence of PSD (43.3%) was noted in

the cross-sectional study conducted in Ethiopia among patients who had a stroke six months or more [37]. However, this high PSD prevalence may be explained by the study sample having an inflated number of patients with moderate (25.8%) to severe (52.6%) strokes [37]. According to another study conducted in Nigeria [29], patients who had a stroke for one year or more had a high prevalence of PSD (80.0%), with 57.14% reporting severe depression [29]. The findings of the study in Nigeria were consistent with the study in Ethiopia in that 98.48% of the study sample had some degree of dependence, with 30.30% reporting partial dependence and 53.03% reporting very and total dependence [29]. Since these two published studies were cross-sectional [29, 37], it is difficult to infer precise causal relationship. In addition, the size and location of brain lesions affecting neurological recovery process are correlated with severe stroke, which could also influence the likelihood of depression [5, 6, 49]. An overview of the PSD prevalence from published studies concluded that the prevalence of PSD varies with the time since a stroke occurs. Peak prevalence typically occurs one to three months following a stroke; however, depression could remain one to five or more years after the incidence of stroke. Persistent or incident depression after a long-term stroke may result from a long-term impairment.

Recovery time after a stroke varies for each patient ranging from several weeks, months, or even years. The most rapid and optimum recovery usually happens in the first three to six months following a stroke [11, 12, 50]. The golden period for stroke recovery is in the first three months after a stroke [50]. Some stroke survivors continue to recover in 12 or 18 months, and their disabled states become permanent afterward [11, 12]. Depression may occur or worse due to this permanent and long-term disability [21, 29], and prior studies indicated that depression is associated with poor quality of life [51, 52]. Since stroke recovery is time-dependent, any factors impeding or slowing down the rehabilitation process must be considered and properly managed.

The results of the systematic review indicate the burden of PSD on functional recovery following a stroke. Depression has an impact on the rehabilitation process, resulting in poor recovery and functional outcomes [14, 15, 17, 20, 21]. In this systematic review, we determined the effects of PSD on functional outcomes after a stroke by four functional outcome measure domains: degree of disability measure, basic activity of daily living measures, instrumental activity of daily living measures, and physical and cognitive function measure.

This study indicates that PSD negatively impacts patients' degree of dependence. The odds of having poor functional independence for PSD patients compared to non-PSD patients were about 1.05 to 2.59 times at three months [15, 17], 2.30 to 3.97 times at one year [15, 21], and 1.80 to 2.28 times at five years [14, 41]. It was known that improvement in the functional outcomes was influenced by the severity and duration of a stroke [5, 53]. The significant impacts of PSD on the degree of dependence were identified among 5 cohort studies with adjustment of baseline stroke severity (NIHSS score) [14, 15, 17, 33, 41]. However, the longitudinal study conducted in South Korea reported that, by 12 months, the majority of chronic stroke patients had permanent disabilities and showed limited independence, regardless of the presence of PSD [33]. It is consistent with previous research suggesting that permanent and long-term disability could induce the onset or exacerbation of depression [21, 29].

Regarding basic activities of daily living (BADL), PSD had a negative association with BADL performance of stroke patients [16, 18, 19, 35]. The cohort study in UK revealed that PSD at three months significantly increased the risk of moderate and severe disability in BADL at three years about 1.89 and 4.01 times, respectively [19]. However, the association between PSD and severe BADL impairment became insignificant after adjusting variables of physical disability, cognitive performance, smoking, use of antidepressant drugs, and social support three months post-stroke [19]. These findings suggest that the functional outcomes of

stroke could be influenced by other potential factors. Nevertheless, depression could be considered one of the important predictors of BADL performance, as treating PSD after its occurrence three months post-stroke led to improvements in BADL performance [19]. Another study in China found significant correlations between depression and BADL dependence three years after the stroke (OR = 1.03) after adjusting for age, stroke severity, cognition, and social support [35]. Two cohort studies revealed that depression during the acute phase of stroke significantly decreased BADL performance at one year (β = -0.162) [18] and three years (β = -16.47) [16] when confounding factors were controlled. Furthermore, a case-control study conducted in Italy reported that PSD patients who responded to antidepressant treatment had higher BADL scores at discharge compared to those who did not respond to treatment [39]. This suggests that investigating and managing PSD is essential for improving BADL performance in patients after stroke [8].

In addition, the instrumental activity of daily living measured by instrumental ADL (IADL) measures also found a negative association between PSD and IADL in stroke patients [28, 30, 31, 34, 37]. Two cohort studies reported that more depressive symptoms significantly decreased IADL scores after stroke at three months [34] and three years [31] after controlling for potential confounding factors. A cross-sectional study in Sweden identified a significant association between depression and work and leisure performance among patients after seven years of their stroke (OR 1.10) [28]. However, since half of the study participants from Sweden were older persons (median age 64), variations in functional performance could potentially be attributed to both depression and other age-related health issues [28].

Regarding physical and cognitive function, the association between PSD and physical and cognitive function, as measured by the Functional Independence Measure (FIM) measure, varied depending on the time after a stroke and how depression was measured. A cohort study conducted in Turkey found that stroke patients without depression showed a better improvement in their physical and cognitive function one month after discharge [32]. On the other hand, a study in Japan found that higher depressive symptom scores significantly reduced physical and cognitive function scores after three months in a rehabilitation ward [38]. Moreover, another cohort study conducted in Japan [36], which assessed post-stroke depression using both a subjective scale (Self-rating depression scale: SDS) and an objective scale (Montgomery–Åsberg Depression Rating Scale: MADRS-J), revealed that the subjective depression scores had no significant impact on recovery of physical and cognitive function one month after a stroke. At the same time, objective depression scores showed a significant impact on the recovery of physical and cognitive function after controlling confounding factors [36]. The instruments' accuracy should be paid more attention to when detecting depression and its significance in requiring treatment.

This systematic review attempted to make a conclusion on the association of PSD to functional outcomes. Unfortunately, the heterogeneity among the included studies was identified due to the variability of study designs, assessment time, depression assessment tools, functional outcome assessment tools, confounding factors controlled, and statistical analysis in identifying associations between PSD and functional outcomes. Therefore, the pooled estimates could not be reported. However, most study findings indicated significant associations between PSD and poor functional outcomes. Stroke-related comorbidities such as hypertension, diabetes mellitus, atrial fibrillation, and heart failure [4] were considered in measuring associations between PSD and functional outcomes in some included studies [14, 15, 19, 33, 41]. The significant associations between PSD and functional outcomes were still presented after controlling these comorbidities as confounding factors. Moreover, the significant associations between PSD and functional outcomes were identified among the studies controlling significant factors of functional outcomes of stroke such as age [14–16, 18, 19, 21, 29–31, 33–38, 41], stroke

severity [14, 15, 17, 19, 28, 31, 33–35, 37, 38, 41], and cognitive impairments [31, 35–38, 41]. Although the strength of associations between PSD and functional outcomes varied with time after a stroke, characteristics of the study population, study designs, and measurements, all significant findings showed that the associations' direction was the same, that PSD was associated with poor functional outcomes.

There are some limitations to this systematic review that need to be acknowledged. First, all included papers were written in English from four databases. Due to exclusions from other databases and high-quality material published in other languages, the review might be limited. Second, the causal relationships between depression and functional outcomes are undermined because this analysis included all observational studies. Third, the included research drew from a heterogeneous study population; it was difficult to draw a conclusion. The different research designs in selected papers might affect estimates of PSD prevalence and its association with functional outcomes. Besides, a few included studies mainly enrolled stroke patients with depression, resulting in an inflation of depression prevalence and the number of severe stroke patients. Finally, the included studies used a variety of depression and functional outcome measurements, resulting in varying conclusions. However, it is important to note that there are various standardized tools for measuring post-stroke depression and functional outcomes.

## Conclusion

This systematic review reveals the magnitude of the PSD problem and emphasizes the impact of post-stroke depression on functional recovery in stroke patients. Although the results may differ by stroke severity and other potential factors, depression was found to have a significantly negative impact on the functional outcomes in stroke patients. Furthermore, this review uncovered that different depression assessment instruments showed noticeably diverse results. Therefore, healthcare professionals need to be adequately trained in the selection and usage of depression assessment tools. Depression is challenging to detect and easily missed since the symptoms are not obvious or concrete. Health professionals who provide direct care for stroke patients should pay attention to post-stroke depression occurrence. Post-stroke patients should be assessed for depression regularly, especially in the first three months after a stroke, which is the golden period for stroke recovery. Prompt PSD management delivery is necessary to promote the best possible stroke recovery. In addition, it is important to routinely assess functional outcomes in order to determine the progression of stroke rehabilitation and patients' support needs.

## Supporting information

**S1 Table. Search strategies.**
(DOCX)

**S2 Table. A summary of the included studies.**
(DOCX)

**S3 Table. The methodological quality of the included studies.**
(DOCX)

**S1 Appendix. PRISMA main checklist.**
(PDF)

**S2 Appendix. PRISMA abstract checklist.**
(PDF)

## Author Contributions

**Conceptualization:** Nipaporn Butsing, Jaclene A. Zauszniewski, Suebsarn Ruksakulpiwat, Atsadaporn Niyomyart.

**Data curation:** Nipaporn Butsing, Atsadaporn Niyomyart.

**Formal analysis:** Nipaporn Butsing, Jaclene A. Zauszniewski, Suebsarn Ruksakulpiwat, Atsadaporn Niyomyart.

**Investigation:** Nipaporn Butsing, Jaclene A. Zauszniewski, Suebsarn Ruksakulpiwat, Atsadaporn Niyomyart.

**Methodology:** Nipaporn Butsing, Jaclene A. Zauszniewski, Suebsarn Ruksakulpiwat, Atsadaporn Niyomyart.

**Project administration:** Nipaporn Butsing, Jaclene A. Zauszniewski, Atsadaporn Niyomyart.

**Resources:** Nipaporn Butsing, Jaclene A. Zauszniewski, Suebsarn Ruksakulpiwat.

**Software:** Nipaporn Butsing.

**Supervision:** Nipaporn Butsing, Jaclene A. Zauszniewski, Mary T. Quinn Griffin, Atsadaporn Niyomyart.

**Validation:** Nipaporn Butsing, Jaclene A. Zauszniewski, Suebsarn Ruksakulpiwat, Mary T. Quinn Griffin, Atsadaporn Niyomyart.

**Visualization:** Nipaporn Butsing, Jaclene A. Zauszniewski, Suebsarn Ruksakulpiwat, Mary T. Quinn Griffin, Atsadaporn Niyomyart.

**Writing – original draft:** Nipaporn Butsing, Jaclene A. Zauszniewski, Atsadaporn Niyomyart.

**Writing – review & editing:** Nipaporn Butsing, Jaclene A. Zauszniewski, Suebsarn Ruksakulpiwat, Mary T. Quinn Griffin, Atsadaporn Niyomyart.

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
