## [Decision Letter · Decision Letter 0]

18 Jul 2024

PONE-D-24-26302Association between post-stroke depression and functional outcomes: A systematic reviewPLOS ONE

Dear Dr. Butsing,

Thank you for submitting your manuscript to PLOS ONE. After careful consideration, we feel that it has merit but does not fully meet PLOS ONE’s publication criteria as it currently stands. Therefore, we invite you to submit a revised version of the manuscript that addresses the points raised during the review process.

We look forward to receiving your revised manuscript.

Kind regards,

Masaki Mogi

Academic Editor

PLOS ONE

Journal Requirements:

**Additional Editor Comments:**

Minor revisions are necessary in the present form.

See the Reviewers' suggestions and respond them appropriately.

Reviewers' comments:

Reviewer's Responses to Questions

**Comments to the Author**

1. Is the manuscript technically sound, and do the data support the conclusions?

Reviewer #1: Yes

Reviewer #2: Yes

2. Has the statistical analysis been performed appropriately and rigorously? 

Reviewer #1: Yes

Reviewer #2: N/A

3. Have the authors made all data underlying the findings in their manuscript fully available?

Reviewer #1: Yes

Reviewer #2: Yes

4. Is the manuscript presented in an intelligible fashion and written in standard English?

Reviewer #1: Yes

Reviewer #2: Yes

5. Review Comments to the Author

Reviewer #1: Reviewer comments re: PONE-D-24-26302

In this study, the authors performed a systematic review and on the association between post-stroke depression and functional outcomes. This is a very important and relevant topic. The study is well-designed and performed.

I feel however, there are some issues that need to be addressed before publication. I have included some suggestions for how the paper could be strengthened.

1. An assessment of the risk of bias in observational epidemiological studies in the original studies was omitted. Review authors should categorize risk of bias as low, medium, or high. It would have been important for the review authors to assess the risk of bias among the included studies.

2. No assessment of publication bias among the included studies is presented in the manuscript. It would have been important for the review authors to at least create a scatter (i.e., funnel) plot of the studies plotting precision (standard error) or sample size by the effect size (mean difference) to assess for publication bias.

3. Another concern is a lot of work by the review authors went into performing this systematic review in an attempt to answer the important question about whether or not post-stroke depression is truly associated with functional outcomes. Unfortunately, the blend of different observational study designs most likely led to significant heterogeneity among the studies, due to variability rather than from random chance alone. Based on the finding of this review, please clarify how the review authors concluded that post-stroke depression is significantly associated with poor functional outcomes, rather than attributing these outcomes directly to the stroke itself? Additionally, to what extent does comorbidity influence patients’ functional outcomes?

Reviewer #2: The review was done systematically.

Please elaborate in the discussion the importance of the findings, and the way forward.

In the limitation, please discuss the different research designs in the selected papers, and how this may affect the prevalence and factors associated.

6. PLOS authors have the option to publish the peer review history of their article (what does this mean?). If published, this will include your full peer review and any attached files.

Reviewer #1: No

Reviewer #2: No

---

## [Author Response · Author response to Decision Letter 0]

3 Aug 2024

Reviewer 1’s Comments

1. An assessment of the risk of bias in observational epidemiological studies in the original studies was omitted. Review authors should categorize risk of bias as low, medium, or high. It would have been important for the review authors to assess the risk of bias among the included studies.

Authors’ Responses: Thank you for your suggestion.

In this systematic review, we used Joanna Briggs Institute (JBI) critical appraisal tools for analytical cohort, cross-sectional, and case-control studies to assess the risk of bias among included studies as mentioned in the Methods and Results sections, and details of the assessment are provided in supplementary material S3 Table. Moreover, we added more explanation of methodological quality as follows:

In Methods section

- Assessment of methodological quality (page 7; line 151).

- Studies with a JBI score higher than 70% were categorized as high quality, moderate quality if scored between 50% and 70%, and low quality if scored lower than 50% (page 7; lines 157-159). 

In Results section

- Methodological quality assessment of the included studies which explained that twenty (95.2%) studies were classified as high quality (low risk of bias), one (4.8%) study was classified as moderate quality (unclear risk of bias). No study with low quality (high risk of bias) was included in this review (pages 11-12; lines 222, 227-229). 

Moreover, the assessment of methodological quality was also added in Table 4 (pages 19-21).

2. No assessment of publication bias among the included studies is presented in the manuscript. It would have been important for the review authors to at least create a scatter (i.e., funnel) plot of the studies plotting precision (standard error) or sample size by the effect size (mean difference) to assess for publication bias.

Authors’ Responses: Thank you for your suggestion.

Publication bias in the form of a scatter plot could not be provided for this review due to significant variations in study design, timing of assessment, depression assessment tools, functional outcome measures, and form of reporting depression and functional outcomes among included papers. Moreover, PSD was the factor that could not be assigned to participants, so it was difficult to identify the case and control group. Although some studies reported sample size and means of PSD and non-PSD participants, they used different tools in measuring functional outcomes, and only two papers reported the same things in that they used the same assessment measures and times. These did not allow us to create a scatter plot of the studies for publication bias assessment. Therefore, we added more explanation and discussion on these limitations in the Methods and Results sections.

The Methods section (page 8; lines 172-173) explained that data synthesis was done narratively since there was heterogeneity in included study characteristics. 

In the Results section, details are shown as follows: 

- Degree of dependence using mRS (page 15; line 278-280) explained that all included studies that assessed mRS reported significant variation in terms of timing of assessment (discharge, 3 months, 12 months, and five years), depression assessment tools (6 different tools), form of report degree of dependence (means or binary grouping). Therefore, these results are presented narratively. 

- Basic ADL (page 16; line 301-303) added that included studies assessed ADL reported significant variation in terms of study designs (cohort, cross-sectional, and case-control), timing of assessment (discharge, 6 weeks, 3 months, 12 months, 2 years, 3 years, five years, and not specific time point), depression assessment tools (7 different tools), form of report ADL (means or groups). Therefore, studies within this group are reported narratively. 

- IADL (pages 16-17; line 322-324) explained that included studies assessed IADL reported significant variation in terms of study designs (cohort, cross-sectional), timing of assessment (3 months, 2 years, seven years, and unidentified), depression assessment tools (5 different tools), form of report IADL (means or groups). Therefore, these results are presented narratively. 

- Physical and cognitive function (page 17; line 342-344), there were only three studies that utilized FIM measurement to measure this functional outcome and these three papers had significant variation in terms of study designs (cohort, cross-sectional), timing of assessment (3 months, 2 years, seven years, and unidentified), depression measures (HADS-D, BDI, PHQ-2, CES-D), reporting of physical and cognitive function. Studies within this group were therefore narratively presented.

3. Another concern is a lot of work by the review authors went into performing this systematic review in an attempt to answer the important question about whether or not post-stroke depression is truly associated with functional outcomes. Unfortunately, the blend of different observational study designs most likely led to significant heterogeneity among the studies, due to variability rather than from random chance alone. Based on the finding of this review, please clarify how the review authors concluded that post-stroke depression is significantly associated with poor functional outcomes, rather than attributing these outcomes directly to the stroke itself? Additionally, to what extent does comorbidity influence patients’ functional outcomes?

Authors’ Responses: Thank you for your suggestion.

We have discussed more about review results and made conclusions based on review findings and limitations as suggested (page 26; lines 455-471).

Reviewer 2’s Comments

Please elaborate in the discussion the importance of the findings, and the way forward.

In the limitation, please discuss the different research designs in the selected papers, and how this may affect the prevalence and factors associated.

Authors’ Responses: Thank you for your suggestion.

The importance of the findings was elaborated in the Discussion section (page 26; lines 455-471). Moreover, the limitations of selected papers were discussed and explained more as suggested. (page 26; lines 455-471, page 27; lines 477-479).

---

## [Decision Letter · Decision Letter 1]

7 Aug 2024

Association between post-stroke depression and functional outcomes: A systematic review

PONE-D-24-26302R1

Dear Dr. Butsing,

We’re pleased to inform you that your manuscript has been judged scientifically suitable for publication and will be formally accepted for publication once it meets all outstanding technical requirements.

Kind regards,

Masaki Mogi

Academic Editor

PLOS ONE

Additional Editor Comments (optional):

Reviewers' comments:

Reviewer's Responses to Questions

**Comments to the Author**

1. If the authors have adequately addressed your comments raised in a previous round of review and you feel that this manuscript is now acceptable for publication, you may indicate that here to bypass the “Comments to the Author” section, enter your conflict of interest statement in the “Confidential to Editor” section, and submit your "Accept" recommendation.

Reviewer #1: All comments have been addressed

Reviewer #2: All comments have been addressed

2. Is the manuscript technically sound, and do the data support the conclusions?

Reviewer #1: Yes

Reviewer #2: Yes

3. Has the statistical analysis been performed appropriately and rigorously? 

Reviewer #1: Yes

Reviewer #2: N/A

4. Have the authors made all data underlying the findings in their manuscript fully available?

Reviewer #1: Yes

Reviewer #2: Yes

5. Is the manuscript presented in an intelligible fashion and written in standard English?

Reviewer #1: Yes

Reviewer #2: Yes

6. Review Comments to the Author

Reviewer #1: (No Response)

Reviewer #2: (No Response)

7. PLOS authors have the option to publish the peer review history of their article (what does this mean?). If published, this will include your full peer review and any attached files.

Reviewer #1: No

Reviewer #2: No

---

## [Editor Report · Acceptance letter]

13 Aug 2024

PONE-D-24-26302R1 

PLOS ONE

Dear Dr. Butsing, 

I'm pleased to inform you that your manuscript has been deemed suitable for publication in PLOS ONE. Congratulations! Your manuscript is now being handed over to our production team.

Kind regards, 

on behalf of

Dr. Masaki Mogi 

Academic Editor

PLOS ONE